# The influence of smartphone addiction on sleep quality among college students: The parallel mediating roles of perceived stress and health-promoting lifestyle

Yuchen Xie[1☯], Qichuan Pei[2☯], Yixiao Chen[2], Lishun Xiao[2], Dehui Yin[2*]

**1** Xuzhou Medical University, Xuzhou, Jiangsu, China, **2** School of Public Health, Xuzhou Medical University, Xuzhou, China

☯ These authors contributed equally to this work.
* yindh16@xzhmu.edu.cn

## Abstract

### Objectives

This study investigated the relationship between smartphone addiction and sleep quality among college students, focusing on the parallel mediating roles of perceived stress and health-promoting lifestyle.

### Methods

A cross-sectional survey was conducted in March 2025 among 2,317 students from Xuzhou Medical University using an online questionnaire. Data were collected via questionnaires and analyzed using SPSS 21.0. The study used the Smartphone Addiction Scale-Short Version (SAS-SV), the Pittsburgh Sleep Quality Index (PSQI), the Health-Promoting Lifestyle Profile (HPLP-II), and Perceived Stress Scale (PSS). Statistical methods included normality tests, descriptive statistics, and mediation analysis.

### Results

A prevalence rate of 51.9% for sleep disorders was identified among the university student population. A statistically significant positive correlation was observed between smartphone addiction and poor sleep quality ($r = 0.259$, $p < 0.01$). Additionally, perceived stress ($r = 0.408$, $p < 0.01$) and health-promoting lifestyle ($r = -0.182$, $p < 0.01$) were identified as parallel mediators in this relationship. Mediation analysis indicated a significant total effect of smartphone addiction (SAS-SV) on sleep quality (PSQI) (path $c = 0.0863$, 95% confidence interval (CI) = 0.0730, 0.0995). Furthermore, a significant direct effect of SAS-SV on PSQI was noted (path $c' = 0.0325$, 95% CI = 0.0188, 0.0461). The health-promoting lifestyle (HPLP) (path $a1b1 = 0.0128$,

**Data availability statement:** Yes - all data are fully available without restriction; All relevant data are within the paper and its Supporting information files.

**Funding:** This work was supported by Jiangsu Province Education Science "14th Five-Year Plan" Planning Projects (JS/2021/GH0106-07330). The funders had no role in study design, data collection and analysis, decision to publish, or preparation of the manuscript.

**Competing interests:** The authors have declared that no competing interests exist.

**Abbreviations:** PSQI, Pittsburgh Sleep Quality Index; SAS-SV, Smartphone Addiction Scale-Short Version; PSS, Perceived Stress Scale; HPLP, Health-Promoting Lifestyle Profile; BMI, body mass index; PA, Physical activity; HR, health responsibility; SM, stress management; NU, nutrition; HR, interpersonal relationship; HC, health care; CI, confidence interval.

95% CI = 0.0086, 0.0176) and perceived stress (PSS) (path a2b2 = 0.0410, 95% CI = 0.0332, 0.0491) were found to partially mediate the relationship between SAS-SV and PSQI, accounting for 14.83% and 47.51% of the total effect, respectively. These findings highlight the dual mediating roles of perceived stress and health-promoting lifestyle in the association between smartphone addiction and sleep quality.

## Conclusions

Smartphone addiction negatively affects sleep quality, both directly and indirectly through increased perceived stress and reduced engagement in a health-promoting lifestyle. Interventions targeting stress management and healthy behaviors are recommended to mitigate these effects.

## Introduction

Sleep is a critical physiological and psychological process that is crucial for supporting numerous essential life functions. It occupies a central position in physiological recovery, as well as in the integration and consolidation of memory. Adequate and restorative sleep serves as a fundamental foundation for maintaining an individual's physical and mental health, in addition to supporting social functioning [1]. Chronic sleep disturbances exert detrimental effects on cognitive performance, diminish vitality, and elevate the risk of accidents. These disturbances are linked to stress, anxiety, depression, mental disorders, mobile phone addiction, physiological diseases, and an increased risk of suicide [2,3]. Given its substantial impact on human health, issues related to sleep health are garnering increasing attention [4].

In recent years, the prevalence of sleep disturbances and mental health issues among college students has become increasingly pronounced, with depression and anxiety being particularly widespread. Prolonged experiences of depression or anxiety can have detrimental effects on students' physiological and psychological health, in addition to their performance in academic and daily functioning. Academic research indicates that sleep quality serves as a pivotal factor influencing the psychological well-being of the collegiate population [5]. Additionally, inadequate proactive health behaviors, mobile phone addiction, depression, and anxiety are significant risk factors that contribute to suboptimal sleep quality [6].

In the contemporary landscape characterized by advanced information technology, the swift evolution of the internet, smartphones, and various forms of media has rendered mobile phones a While these devices significantly enhance modern living and empower individuals, their excessive use or addiction has precipitated a range of troubling issues. Compared with other population groups, college students demonstrate a heightened vulnerability to mobile phone addiction, as evidenced by existing studies. [7,8]. Numerous studies have substantiated the deleterious impact of smartphone addiction on the nocturnal rest of college students [9,10].

Stress has emerged as a pressing public health issue in contemporary society. Research indicates a bidirectional relationship between stress and mobile phone

usage. For example, some students may utilize mobile phones as a means to mitigate stress, whereas others who excessively engage with their devices may encounter increased stress levels due to perpetual connectivity with peers [11]. Furthermore, psychological stress among students can result in a range of sleep disturbances, including restless sleep, nocturnal awakenings, and early morning arousals [12,13].

Health-promoting behaviors are characterized as lifestyle choices that individuals adopt to improve their health, well-being, and self-fulfillment [14]. These behaviors encompass a range of multidimensional, spontaneous, and sustained daily activities, including the maintenance of a balanced diet and the engagement in regular physical exercise [15]. Empirical research has consistently indicated that health-promoting behaviors are instrumental in influencing sleep quality, just as unhealthy lifestyle patterns pose a significant detriment to sleep integrity [16]. For example, the adoption of a healthy lifestyle, which includes the consumption of nutritious foods and regular physical activity, has been empirically demonstrated to exert a beneficial effect on sleep quality [17]. Nevertheless, the majority of existing investigations have predominantly concentrated on the detrimental effects of inappropriate behaviors, such as sedentary lifestyles, alcohol consumption, and smoking, on sleep quality [18,19]. Consequently, there has been limited exploration of the relationship between various forms of health-promoting behaviors and sleep outcomes.

It is noteworthy that there is a limited number of studies that have thoroughly investigated the influence of smartphone usage on health-promoting behaviors and their subsequent effects on sleep quality [17]. This deficiency in the existing literature highlights the necessity for a more comprehensive approach to understanding how smartphone dependence may disrupt health-promoting behaviors and, consequently, impact sleep quality. By systematically exploring the interactions among health-promoting behaviors, smartphone usage, and sleep quality, significant insights can be obtained that may assist college students in enhancing their sleep and overall well-being.

Consequently, based on the aforementioned literature review and theoretical frameworks, we formulated the hypotheses for our hypothetical model (Fig 1). The Smartphone Addiction Scale-Short Version (SAS-SV) demonstrated positive correlations with the Pittsburgh Sleep Quality Index (PSQI) and the (Perceived Stress Scale) PSS, and a negative correlation with the Health-Promoting Lifestyle Profile (HPLP). We further hypothesize that the HPLP exhibited a negative correlation with the PSQI, whereas the PSS exhibited a positive correlation with it.

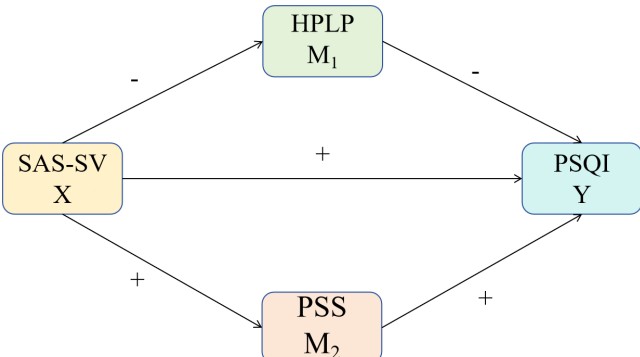

**Fig 1. Hypothetical model.** A mediation analysis was conducted with the SAS-SV as the independent variable (X), the PSQI as the dependent variable (Y), and the PSS and the HPLP-II as the mediators M1 and M2. The coefficient c is the total effect between X and Y, and c′ is the direct effect of X on Y whilst controlling for M1 and M2. The SAS-SV demonstrated positive correlations with the PSQI and the PSS, and a negative correlation with the HPLP. We further hypothesize that The HPLP exhibited a negative correlation with the PSQI, whereas the Perceived Stress Scale (PSS) exhibited a positive correlation with it.

## Methods

### Ethics approval and consent to participate

The study was conducted in accordance with the Declaration of Helsinki, and approved by the Ethics Committee of Xuzhou Medical University (approval no.: XZHMU-2024G062). The participants provided their informed consent to participate in this study.

### Participants

In March 2025, an online questionnaire survey was conducted utilizing the Questionnaire Star platform among students at Xuzhou Medical University, employing a random cluster sampling method. The survey was anonymous. We did not collect any identifying information during or after data collection. Initially, the study included 2,330 participants. However, after excluding 13 invalid questionnaires due to criteria such as outliers in height and weight and incomplete responses, the final analytical sample comprised a total of 2,317 participants, resulting in a questionnaire validity rate of 99.44%. The survey collected respondents' basic information, which included gender, grade, major, place of birth, cost of living, whether they were only children, body mass index (BMI), and duration of smartphone usage.

### Measurement instruments

The present study utilized a questionnaire survey to examine students' dependence on smartphones, sleep quality, stress levels, and health-promoting lifestyles. The primary components of the survey encompassed:

**Smartphone addiction scale-short version.** The SAS-SV [20] is a validated self-report instrument designed to assess smartphone addiction. This scale has been extensively validated across rigorous empirical studies involving collegiate students from diverse linguistic and cultural contexts [21–23]. The SAS-SV consists of 10 items, each rated on a six-point Likert scale ranging from 1 ("strongly disagree") to 6 ("strongly agree"). The total score, derived from the summation of all item scores, serves as an indicator of the severity of smartphone addiction, with higher scores reflecting a greater propensity for addictive behavior. In the present study, the internal consistency of the SAS-SV, as measured by Cronbach's α, was found to be 0.877, indicating high reliability.

**Pittsburgh sleep quality index.** The quality of sleep was evaluated utilizing the PSQI [24]. This comprehensive questionnaire comprises 19 items organized into seven distinct dimensions: subjective sleep quality, sleep latency, sleep duration, habitual sleep efficiency, sleep disturbances, use of sleep medications, and daytime dysfunction. Each dimension is scored on a scale from 0 to 3, culminating in a global score that ranges from 0 to 21. Notably, higher global scores are indicative of poorer sleep quality. In the current study, the PSQI demonstrated strong internal consistency, with a Cronbach's alpha coefficient of 0.855. Consistent with prior research, a global PSQI score exceeding 5 was classified as indicative of "poor sleep quality."

**Perceived Stress Scale.** The PSS-10 [25] comprises ten items constructed to assess individuals' stress levels over the preceding month. Participants self-report their responses on a straightforward five-point Likert scale, ranging from 0 ("never") to 4 ("very often"). Items 1, 2, 3, 6, 9, and 10 are scored directly (0 = "never" to 4 = "very often"), while items 4, 5, 7, and 8 are reverse-scored (0 = "very often" to 4 = "never") [26]. The total score is derived by aggregating the responses to all ten items, producing a range from 0 to 40, with higher scores indicating greater perceived stress. In this study, the PSS-10 demonstrated a Cronbach's α of 0.707, indicating acceptable internal consistency.

**Health-promoting lifestyle profile-II.** Pender et al. [27] developed the HPLP in 1987, which contains 48 items in six dimensions, and Walker et al. [28] revised the HPLP in 1995 to form the HPLP-II, which has 52 items in six dimensions. Physical activity (PA), health responsibility (HR), stress management (SM), nutrition (NU), interpersonal relationship (IR), and health care (HC). In this study, the Cronbach's alpha of HPLP-II was 0.965.

## Statistical analysis

Descriptive statistical analyses, including t-tests and Pearson's correlation analyses, were conducted using IBM SPSS version 21.0. All measurements were tested for normality and were found to follow a non-normal distribution. The median (M) and interquartile range (P25, P75) served to characterize the scale scores due to their non-normal distribution, as indicated by the Kolmogorov-Smirnov test ($P < 0.05$). Pearson's correlation was employed to investigate the relationships among cell phone addiction, psychological resilience, and sleep quality. The mediating effect of psychological resilience was tested using Model 4.2 (parallel mediation) of the PROCESS macro version 4.1 in SPSS 21.0. Additionally, 95% confidence intervals (CIs) were estimated through bootstrapping with 5000 resamples to evaluate the indirect effects of each variable. An indirect effect was deemed significant if the 95% CI did not include 0. Statistical significance was established at $P < 0.05$ (two-tailed).

## Results

### Harman's single-factor test

Utilizing Harman's single-factor test method, all items were computed and analyzed. The findings demonstrated that 15 factors emerged with eigenvalues exceeding 1, with the highest factor variance explanation rate being 23.454%, which did not surpass the threshold of 40% [29]. Consequently, no statistically significant common method bias was detected in this study.

### Characterization analysis

This study included a total of 2,317 participants, consisting of 800 males (34.53%) and 1,517 females (65.47%). The majority of participants were medical students, accounting for 76.48% of the sample. In terms of academic classification, there were 1,315 freshmen, 472 sophomores, 348 juniors, and 182 seniors or individuals in higher academic years. Among the participants, 937 were only children, while 1,380 were not. Sleep disorders were reported in 375 males (46.88%) and 735 females (48.45%). Notably, 57.69% of senior medical students and those in higher academic years experienced sleep disorders. Furthermore, nearly half of the students (47.7%) reported using their smartphones for five hours or more daily. Overall, 51.9% (1,110 out of 2317) of all participants exhibited sleep disturbances, as indicated by a PSQI score greater than 5. As indicated in Table 1, gender, academic year, being an only child, and the duration of smartphone usage were identified as significant factors influencing sleep quality ($P < 0.05$).

### Scores on various scales for college students with diverse characteristics

The scores from the four scales-SAS-SV, PSQI and PSS were analyzed with respect to various demographic attributes, including gender, grade level, and academic major, serving as grouping factors. The results of the intergroup comparisons indicated that male students demonstrated significantly higher scores for mobile phone addiction in comparison to their female counterparts. Moreover, the SAS-SV scale revealed statistically significant differences in mobile phone addiction scores across the four grade levels. Additionally, a significant positive association was identified between the duration of mobile phone usage and the scores on the SAS-SV, PSQI, and PSS scales. This finding suggests that prolonged mobile phone use bears a relationship to elevated levels of stress perception, diminished sleep quality, and an increase in intuitive eating tendencies. Comprehensive results are provided in Table 2.

### Correlation among the primary study variables

The Pearson product-moment correlation analysis, as presented in Table 3, indicated significant positive correlations between the SAS-SV and the PSQI ($r = 0.259$, $p < 0.01$), as well as between SAS-SV and the PSS-10 ($r = 0.408$, $p < 0.01$). In contrast, SAS-SV demonstrated a statistically significant inverse association with the HPLP ($r = -0.182$, $p < 0.01$).

**Table 1. Characterization of the participants for the full samples (N=2317).**

| Group | | PSQI≤5 | PSQI>5 | $X^2$ | P | N(%) |
|---|---|---|---|---|---|---|
| Gender | | | | 55.392 | <0.001 | |
| | Male | 425 | 375 | | | 800 (34.53%) |
| | Female | 782 | 735 | | | 1517 (65.47%) |
| Major | | | | 2.748 | 0.106 | |
| | Medical | 940 | 832 | | | 1772 (76.48%) |
| | Non-medical | 267 | 278 | | | 545 (23.52%) |
| Grade | | | | 7.956 | <0.001 | |
| | First-year | 666 | 649 | | | 1315 (56.75%) |
| | Second-year | 285 | 187 | | | 472 (20.37%) |
| | Third-year | 179 | 169 | | | 348 (15.02%) |
| | Fourth-year or more | 77 | 105 | | | 182 (7.86%) |
| Native place | | | | 2.506 | 0.114 | |
| | Urban | 556 | 486 | | | 1042 (44.97%) |
| | Rural | 651 | 624 | | | 1275 (55.03%) |
| BMI | | | | 0.079 | 0.972 | |
| | ≤18.5 | 152 | 149 | | | 301 (12.99%) |
| | 18.5-24 | 717 | 654 | | | 1371 (59.17%) |
| | 24-28 | 221 | 199 | | | 420 (18.13%) |
| | >28 | 117 | 108 | | | 225 (9.71%) |
| Only child or not | | | | 2.613 | 0.016 | |
| | Yes | 506 | 431 | | | 937 (40.44%) |
| | NO | 701 | 679 | | | 1380 (59.56%) |
| Cost of living (yuan) | | | | 1.649 | 0.176 | |
| | <800 | 13 | 20 | | | 33 (1.42%) |
| | 800-1600 | 454 | 409 | | | 863 (37.25%) |
| | 1600-2400 | 632 | 574 | | | 1206 (52.05%) |
| | >2400 | 108 | 107 | | | 215 (9.28%) |
| Daily smartphone usage duration | | | | 17.360 | <0.001 | |
| | <1h | 32 | 7 | | | 39 (1.68%) |
| | 1-3h | 168 | 119 | | | 283 (12.21%) |
| | 3-5h | 488 | 402 | | | 890 (38.41%) |
| | 5-8h | 405 | 410 | | | 815 (35.17%) |
| | >8h | 114 | 176 | | | 90 (12.53%) |

Furthermore, a positive correlation was found between PSS-10 and PSQI (r=0.366, $p<0.01$), while a statistically significant negative correlation was identified between HPLP and PSQI (r=−0.213, $p<0.01$).

## The mediating effect of HPLP and PSS

We utilized Preacher and Hayes' parallel mediation model (Model 4.2) [30] with 95% CI derived from 5,000 bootstrap resamples to investigate the indirect effects of the SAS-SV on the PSQI through the HPLP and PSS. A mediation analysis was conducted with the SAS-SV as the independent variable (X), the PSQI as the dependent variable (Y), and the PSS and the HPLP-II as the mediators M1 and M2, respectively. Consistent with the guidelines provided by Becker et al. [31], we present the results without the inclusion of control variables. The outcomes of the final mediation analysis are illustrated in Fig 2 and detailed in Table 4 [32]. The results indicated a significant total effect of SAS-SV on PSQI

**Table 2. Scores on various scales for college students with different characteristics.**

| Group | SAS-SV | | PSQI | | PSS | | HPLP | |
|---|---|---|---|---|---|---|---|---|
| | M ($P_{25}$, $P_{75}$) | Z/H | M ($P_{25}$, $P_{75}$) | Z/H | M ($P_{25}$, $P_{75}$) | Z/H | M ($P_{25}$, $P_{75}$) | Z/H |
| Gender | | −6.143*** | | −1.632 | | −3.232*** | | −2.386* |
| Male | 32 (**25,38**) | | 5 (3,8) | | 19 (16,21) | | 133 (115,154) | |
| Female | 34 (**29,39**) | | 5 (4,7) | | 20 (17,22) | | 152 (136,166) | |
| Major | | −0.337 | | −2.406* | | −1.738 | | −2.57*** |
| Medical | 34 (27.25,39) | | 5 (3,7) | | 20 (17,21) | | 136 (119,153) | |
| Non-medical | 34 (28,39) | | 6 (4,8) | | 20 (17,22) | | 133 (116,151) | |
| Grade | | 9.166* | | 25.196*** | | 11.826** | | 11.795*** |
| First-year | 33 (**27,38**) | | 5 (4,7) | | 20 (17,21) | | 137 (120,154) | |
| Second-year | 34 (**27,39**) | | 5 (3,7) | | 19 (17,21) | | 137 (117,154) | |
| Third-year | 35 (**29,40**) | | 5 (4,7) | | 20 (17,22) | | 129 (116,147) | |
| Fourth-year or more | 33.5 (**28,40**) | | 6 (4,8) | | 20 (17.75,23) | | 131 (118,148.25) | |
| Native place | | −2.272* | | −1.544 | | −1.353 | | −3.8*** |
| Urban | 33 (**27,39**) | | 5 (3,7) | | 19 (17,22) | | 138 (120,155) | |
| Rural | 34 (**28,39**) | | 5 (4,7) | | 20 (17,21) | | 133 (117,150) | |
| BMI | | 2.545 | | 0.466 | | 0.132 | | 5.869 |
| ≤18.5 | 33 (27,38) | | 5 (4,7) | | 19 (17,22) | | 132 (116,151) | |
| 18.5-24 | 34 (28,39) | | 5 (4,7) | | 20 (17,21) | | 136 (119,154) | |
| 24-28 | 34 (27,39) | | 5 (3,7) | | 20 (17,21) | | 135 (118,151) | |
| >28 | 34 (27.5,40) | | 5 (3,8) | | 20 (17,22) | | 134 (117,153) | |
| Only child or not | | −3.901*** | | −1.400 | | −1.878 | | −2.923** |
| Yes | 33 (**26,38**) | | 5 (3,7) | | 19 (16.5,21) | | 138 (120,155) | |
| No | 34 (**29,39**) | | 5 (4,7) | | 20 (17,21) | | 134 (118,150) | |
| Cost of living (yuan) | | 1.721 | | 3.695 | | 4.58 | | 26.737*** |
| <800 | 34 (**29,40**) | | 6 (4,9) | | 20 (18,23) | | 126 (104,155) | |
| 800-1600 | 34 (**28,39**) | | 5 (3,7) | | 20 (17,21) | | 132 (117,149) | |
| 1600-2400 | 34 (**28,38.25**) | | 5 (4,7) | | 19 (17,21) | | 136 (120,153.25) | |
| >2400 | 33 (**27,39**) | | 5 (3,7) | | 19 (17,22) | | 141 (124,158) | |
| Daily smartphone usage duration | | 369.297*** | | 71.957*** | | 81.021** | | 30.942*** |
| <1h | 12 (10,22) | | 3 (1,5) | | 12 (0,18) | | 145 (104,201) | |
| 1-3h | 29 (22,34) | | 5 (3,7) | | 18 (16,21) | | 140 (121,156) | |
| 3-5h | 32 (26,37) | | 5 (3,7) | | 19 (17,21) | | 138 (121,153.25) | |
| 5-8h | 35 (31,40) | | 6 (4,8) | | 20 (17,22) | | 132 (118,148) | |
| >8h | 39.5 (34,45) | | 6 (4,9) | | 20 (18,23) | | 130 (113,150.25) | |

M, median; P25, lower quartile; P75, upper quartile. *$P<0.05$,**$P<0.01$,***$P<0.001$.

(path c = 0.0863, 95% CI = 0.0730, 0.0995). Moreover, a significant direct effect of SAS-SV on PSQI was identified (path c′ = 0.0325, 95% CI = 0.0188, 0.0461). Additionally, HPLP (path $a_1b_1$ = 0.0130, 95% CI = 0.0088, 0.0175) and PSS (path $a_2b_2$ = 0.0410, 95% CI = 0.0332 0.0491) were found to partially mediate the relationship between SAS-SV and PSQI, accounting for 14.83% and 47.51% of the total effect of SAS-SV on PSQI, respectively.

## Discussion

This study investigated the influence of smartphone dependence on sleep quality within the university student population, with a particular emphasis on the mediating roles of perceived stress and health-promoting behaviors. The

**Table 3. Correlations between the main study variables (N = 2317).**

|  | SAS-SV | PSS-10 | HPLP | PSQI |
|---|---|---|---|---|
| SAS-SV | 1 |  |  |  |
| PSS-10 | 0.408** | 1 |  |  |
| HPLP | −0.182** | −0.085** | 1 |  |
| PSQI | 0.259** | 0.366** | −0.213** | 1 |

**P<0.01.

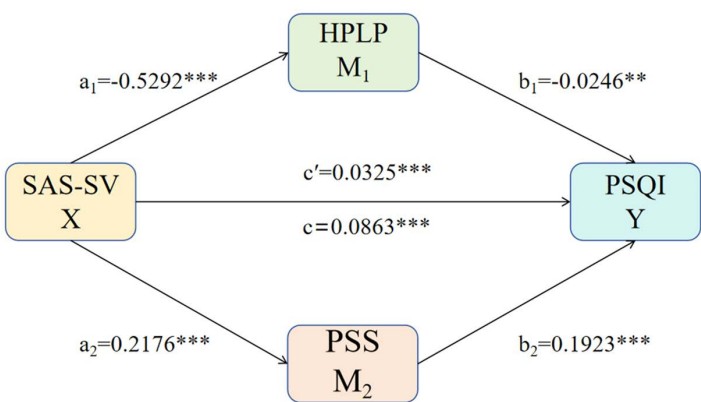

**Fig 2. Mediation model of HPLP and PSS between SAS-SV and PSQI.** The path coefficients in the parallel mediation model are delineated as follows. The coefficient $a_1$ signifies that the SAS-SV is a significant positive predictor of the PSS, whereas $a_2$ denotes its significant negative predictive effect on the HPLP-II. After controlling for SAS-SV and HPLP-II, the path coefficient $b_1$ indicates that the PSS remains a significant positive predictor of the PSQI. Conversely, after controlling for the SAS-SV and PSS, the coefficient $b_2$ shows that the HPLP-II is a significant negative predictor of PSQI. Furthermore, two specific indirect effects were identified: smartphone addiction impairs sleep quality by increasing perceived stress ($a_1b_1$), and similarly, by reducing health-promoting lifestyle behaviors ($a_2b_2$). The coefficient c is the total effect between X and Y, and c′ is the direct effect of X on Y whilst controlling for M1 and M2. ***P<0.001.

**Table 4. Mediation analysis of SAS-SV and PSQI.**

|  | Effect | Boot.SE | Boot 95% CI | | P | Proportion |
|---|---|---|---|---|---|---|
|  |  |  | Lower | Upper |  |  |
| Total effect | 0.0863 | 0.0068 | 0.0730 | 0.0995 | <0.001 | 100% |
| Direct effect | 0.0325 | 0.0069 | 0.0188 | 0.0461 | <0.001 | 37.66% |
| Total Indirect effect | 0.0538 | 0.0040 | 0.0459 | 0.0617 | <0.001 | 62.34% |
| SAS→HPLP→PSQI | 0.0128 | 0.0023 | 0.0086 | 00176 |  | 14.83% |
| SAS→PSS→PSQI | 0.0410 | 0.0041 | 0.0332 | 0.0491 |  | 47.51% |

findings revealed a significant positive correlation between smartphone dependence and poor sleep quality. Additionally, health-promoting behaviors exhibited a negative correlation with both smartphone dependence and sleep quality. Conversely, perceived stress was positively correlated with smartphone dependence and sleep quality. Mediation analysis indicated that smartphone dependence exerted both a direct effect on sleep quality and an indirect effect mediated through perceived stress (indirect effect = 0.0410, 95% CI = 0.0332, 0.0491) and health-promoting behaviors (indirect effect = 0.0128, 95% CI = 0.0086, 0.0176). These results highlight the dual pathways through which smartphone

dependence affects sleep quality, emphasizing the necessity of addressing both perceived stress and health-promoting behaviors in interventions designed to enhance sleep outcomes within the university student population.

While the cross-sectional design precludes definitive causal inferences, the identified parallel mediation model aligns with and can be interpreted through several plausible mechanisms derived from existing literature. The pathway through perceived stress may operate via: (a) physiological arousal from screen-based stimuli and blue light exposure before bedtime, which directly heightens cognitive alertness and delays sleep onset; (b) psychological burdens such as fear of missing out, social comparison, and information overload inherent in sustained social media and messaging app use, contributing to chronic stress that dysregulates the hypothalamic-pituitary-adrenal axis and impairs sleep continuity. Concurrently, the pathway through health-promoting lifestyle may be explained by a displacement effect, wherein excessive smartphone use occupies time and cognitive resources that would otherwise be allocated to activities such as physical exercise, preparatory sleep routines, and face-to-face social interactions—all of which are known to support sleep physiology and psychological resilience. Thus, smartphone addiction likely erodes sleep quality both by directly increasing psychophysiological stress and by crowding out the very behaviors that buffer against stress and promote restorative sleep.

This study elucidates the considerable influence of smartphone addiction on sleep quality within the university student population, with perceived stress and health-promoting lifestyle serving as mediating factors. The findings are in alignment with prior research that has documented the adverse effects of excessive smartphone usage on both mental and physical health [3,17]. The observed positive correlation between smartphone addiction and perceived stress indicates that students who engage in excessive smartphone use may experience elevated levels of stress, potentially attributable to factors such as constant connectivity, social media pressures, or disruptions to their daily routines [33,34]. This heightened stress, in turn, may contribute to sleep disturbances, including difficulties in initiating sleep, frequent awakenings, and overall poor sleep quality [35,36].

Moreover, the inverse relationship between smartphone addiction and a health-promoting lifestyle suggests that excessive use of smartphones may detract from the time and energy that could be allocated to engaging in healthy behaviors, including physical activity, balanced nutrition, and sufficient rest [15,16]. This decline in health-promoting activities intensifies sleep-related issues, thereby establishing a detrimental cycle that further undermines the well-being of students.

It has long been acknowledged that a healthy lifestyle is advantageous for the maintenance and enhancement of health; however, college students frequently engage in unhealthy behaviors, including insufficient sleep, poor dietary choices, and inadequate physical activity [37,38]. Conversely, the adoption of a healthy lifestyle can facilitate the attainment and preservation of optimal physical and mental well-being. For instance, regular physical exercise has been shown to alleviate feelings of fatigue, diminish symptoms of anxiety and depression, and consequently enhance sleep quality [39].

Prior studies have documented a significant association between stress management and sleep quality among college students [40,41]. Numerous studies have identified stress as a significant predictor of sleep quality in this demographic, potentially contributing to sleep deprivation. While the current study, along with previous research, has not explicitly identified the specific types of stress that adversely affect sleep quality within the university student population, the findings from the six HPLP domains in this study suggest that certain stressors may arise from intra-personal functioning and issues related to interpersonal support. Consequently, this finding indicates that promoting the development of self-affirmation, positive attitudes, strong interpersonal relationships, and effective stress management skills among students may enhance their sleep quality more significantly.

The parallel mediation model highlights the dual pathways by which smartphone addiction impacts sleep quality. Addressing a single mediator may prove inadequate in alleviating the detrimental effects of smartphone addiction. Therefore, comprehensive interventions that concurrently target stress reduction and the promotion of healthy lifestyles are essential. For example, stress management programs, mindfulness training, and educational initiatives regarding the advantages of balanced smartphone usage could be incorporated into university curricula. Furthermore, encouraging

students to participate in regular physical activity, adhere to a nutritious diet, and establish consistent sleep routines may mitigate the adverse consequences associated with smartphone addiction.

In conclusion, this study offers significant insights into the intricate relationship between smartphone addiction and sleep quality within the university student population. By identifying perceived stress and health-promoting lifestyle as concurrent mediators, it establishes a framework for the development of targeted interventions aimed at enhancing sleep quality and overall well-being within this demographic. It is imperative that universities and policymakers prioritize strategies that address both the psychological and behavioral ramifications of smartphone addiction to promote healthier lifestyles and improve sleep outcomes among students.

## Limitations

This study presents several limitations. Firstly, the reliance on convenience sampling, with participants exclusively drawn from Xuzhou Medical University, may restrict the generalizability of the findings. Subsequent research should endeavor to adopt broader sampling approaches that encompasses multiple institutions and regions to obtain more representative data. Secondly, the cross-sectional design of this study constrains the capacity to establish causal relationships; therefore, prospective longitudinal studies are necessary to gain deeper insights into the phenomena under investigation. Furthermore, while this study predominantly employed quantitative methods, the integration of qualitative approaches could significantly enhance the depth and richness of future research. Lastly, the data collected were based on self-reports, which are susceptible to subjective bias. Future investigations should consider the use of objective measures, such as ActiGraph or polysomnography, to provide a more accurate understanding of sleep patterns among university students.

## Supporting information

**S1 File. The raw data of the manuscript.**
(XLSX)

## Author contributions

**Conceptualization:** Yuchen Xie, Dehui Yin.

**Formal analysis:** Qichuan Pei, Yixiao Chen.

**Funding acquisition:** Yuchen Xie.

**Methodology:** Qichuan Pei.

**Writing – original draft:** Dehui Yin.

**Writing – review & editing:** Lishun Xiao.

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
