## [Decision Letter · Decision Letter 0]

25 Nov 2025

Dear Dr. Yin,

We look forward to receiving your revised manuscript.

Kind regards,

Pavle Randjelovic, Ph.D.

Academic Editor

PLOS ONE

Journal Requirements:

The impact of smartphone dependence on college students’ sleep quality: the chain-mediated role of negative emotions and health-promoting behaviors - https://doi.org/10.3389/fpubh.2024.1454217

The chain mediating role of family health and physical activity in the relationship between life satisfaction and health-promoting lifestyles among young adults in China - https://doi.org/10.3389/fpubh.2024.1408988

In your revision ensure you cite all your sources (including your own works), and quote or rephrase any duplicated text outside the methods section. Further consideration is dependent on these concerns being addressed.

“This work was supported by Jiangsu Province Education Science "14th Five-Year Plan" Planning Projects (JS/2021/GH0106-07330). The funders had no role in study design, data collection and analysis, decision to publish, or preparation of the manuscript.”

4. In the online submission form, you indicated that [The datasets used and/or analyzed during the current study are available from the corresponding author on reasonable request.].

Reviewers' comments:

Reviewer's Responses to Questions

**Comments to the Author**

1. Is the manuscript technically sound, and do the data support the conclusions?

Reviewer #1: Yes

Reviewer #2: Partly

2. Has the statistical analysis been performed appropriately and rigorously?

Reviewer #1: Yes

Reviewer #2: Yes

3. Have the authors made all data underlying the findings in their manuscript fully available?

Reviewer #1: Yes

Reviewer #2: No

4. Is the manuscript presented in an intelligible fashion and written in standard English?

Reviewer #1: Yes

Reviewer #2: Yes

Reviewer #1: This manuscript is clearly written, well organized, and addresses a relevant public health issue. The study benefits from a solid methodological framework, including the use of validated measurement tools, an appropriate analytical strategy, and a sufficiently large sample to support its conclusions. The results are presented in a straightforward manner and the discussion is well grounded in the existing literature. The findings contribute meaningfully to the understanding of the mechanisms linking smartphone addiction and sleep quality, particularly through perceived stress and health-promoting lifestyle. Overall, the work is scientifically sound and fits well within the scope of PLOS ONE.

Reviewer #2: In the current study, the authors investigated the relationship between smartphone addiction and sleep quality among college students and specifically tested whether perceived stress and health-promoting lifestyle act as parallel mediators. This study employed a cross-sectional design, with data collected at a single time point (March 2025) among 2317 students. The findings revealed a significant positive correlation between smartphone addiction and poor sleep quality. Furthermore, it highlights the dual mediating roles of perceived stress and health-promoting lifestyle as mediators in this relationship. Overall, the study is interesting. However, some points require clarification throughout to improve the manuscript further. I can only think of a few suggestions, as detailed below, that came into my mind.

Comment #1: The abstract (lines 14-50) is long. Please shorten.

Comment #2: Please make sure that you explain all abbreviations when they are first mentioned in the abstract and in the main text. No need to repeat the full meaning in the text unless it is necessary. Also, it would be nice to include all abbreviations in the provided section at 326-329.

Comment #3: I see that the authors present scientific hypotheses at the end of the introduction (lines 109-117) to justify the aim of this research. I think it is better to briefly explain how the study was performed to verify these hypotheses and highlight the main findings of this study, and how the results support the hypotheses. The transition from line 114 to line 115 is not logical. Do you still elaborate on the hypothesis or move to explain your findings? Please review.

Comment #4: In the statistical analysis, two sentences contradict each other. The first statement indicates that all variables are normally distributed (lines 174-175). The second states that the scale scores are not normally distributed, as indicated by the Kolmogorov–Smirnov test (lines 178-179). Pearson’s correlation was used (lines 179–180), although the data were previously described as non-normally distributed. If the scale scores indeed deviate from normality, Spearman’s rank correlation would generally be more appropriate. However, Pearson’s correlation can still be valid with large sample sizes or when deviations from normality are mild. Please clarify this discrepancy and justify the use of Pearson’s correlation.

Comment #5: Very limited information is provided on how the mediation analysis was performed. It refers to the model used for parallel mediation, as Model 4 at line 230, but it also stated as Model 4.2 at line 182. I am wondering whether the parallel mediation model proposed by Preacher and Hayes (2008) (line 230), DOI:10.3758/BRM.40.3.879, using the PROCESS feature (line 182) was used. Then, please include a proper reference. Also, no explanation of the model’s equations is provided (e.g., direct and indirect paths, meaning of c, c′, a₁b₁, a₂b₂ in your model).

Comment #6: At lines 195, 203, 463, I think there is a typo, where the total number is 2,317 (lines 21, 126), not 2,137. Please check.

Comment #7: For mediation analysis and values, I noticed some discrepancies appeared between the text (lines 236-241, Table 4, and Figure 2 regarding the values of the paths. For example, at line 237 and figure 2, the value for path c’ is 0.03145; however, in Table 4 the value is 0.0325. At line 238, the value for path a1b1 is 0.0130, while in Table 4 it is 0.0128. same issue for a2b2. In Table 4, I would prefer to specify the paths corresponding to each row as in Figure 2. Please verify, check and unify the decimal throughout the manuscript.

Comment #8: In Tables 1 and 2, please specify whether the smartphone usage duration is measured per day or in another way. And whether every use is counted, including for learning purposes.

Comment #9: Figure related: The legend of Figure 1 contains very limited information. I would rather expand some information about the abbreviations, meaning of the “+”, “-”, “M1”, “M2”, “X”, “Y”, and briefly explain your logical flow and proposed hypothesis. Same issue for Figure 2, no explanation for direct, indirect paths, meaning a₁, b₁, a₂, b₂.

Comment #10: Given that the results showed that smartphone addiction is associated with poorer sleep quality. And the perceived stress and lifestyle mediate the relationship statistically. Additionally, given the study's limitations, it is not possible to infer causal relationships. However, I am missing information on whether smartphone addiction could lead to poor sleep quality because it increases stress and decreases healthy lifestyles, or what exactly the potential mechanisms for this association.

**Do you want your identity to be public for this peer review?** For information about this choice, including consent withdrawal, please see our Privacy Policy

Reviewer #1: No

Reviewer #2: No

---

## [Author Response · Author response to Decision Letter 1]

15 Dec 2025

Dear Editor and Reviewer,

Thank you for your comments. We have revised our manuscript carefully according to your comments and all the major changes have been marked in red color. We appreciate these useful comments and suggestions from you. The responses are as follows:

Response to Editor,

Response: We have carefully reviewed the provided style template and have reformatted the entire manuscript to ensure it fully complies with PLOS ONE's style requirements, including file naming conventions. All formatting details (e.g., font, line spacing, heading levels, figure placement) have been adjusted accordingly.

The impact of smartphone dependence on college students’ sleep quality: the chain-mediated role of negative emotions and health-promoting behaviors - https://doi.org/10.3389/fpubh.2024.1454217

The chain mediating role of family health and physical activity in the relationship between life satisfaction and health-promoting lifestyles among young adults in China - https://doi.org/10.3389/fpubh.2024.1408988

In your revision ensure you cite all your sources (including your own works), and quote or rephrase any duplicated text outside the methods section. Further consideration is dependent on these concerns being addressed.

Response: We performed plagiarism assessments on the manuscript utilizing Turnitin and thoroughly examined the two articles you supplied. Corresponding revisions have been implemented, resulting in a substantially reduced similarity index, with the exception of the Methods section. Additionally, we have incorporated appropriate citations for all relevant sources where necessary, either by directly quoting or extensively paraphrasing any duplicated content outside the Methods section. The revised manuscript now guarantees that all sources are properly acknowledged and that the text is either original or correctly cited. All modifications have been highlighted for your convenience.

“This work was supported by Jiangsu Province Education Science "14th Five-Year Plan" Planning Projects (JS/2021/GH0106-07330). The funders had no role in study design, data collection and analysis, decision to publish, or preparation of the manuscript.”

Response: As requested, we have amended the Funding Statement to declare all sources of support for this study.

4. In the online submission form, you indicated that [The datasets used and/or analyzed during the current study are available from the corresponding author on reasonable request.].

Response: We have ensured compliance with PLOS’s data policy. All data required to replicate the findings are now freely available as Supporting Information file uploaded with this submission. We have updated the Data Availability Statement in the manuscript to reflect this and revised the online submission form.

Response: We have revised the manuscript as requested: Moved the ethics statement exclusively to the Methods section. Deleted any duplicate ethics statements from all other sections of the manuscript.

Response: Thank you for the suggestions. We have thoroughly evaluated the recommended publications.

Response: We have verified that all references cited in the text are present in the list. All bibliographic details have been checked for accuracy against the original sources. We have specifically screened the entire list against databases and confirmed that no cited articles have been retracted.

Response to Reviewer 2

Comment #1: The abstract (lines 14-50) is long. Please shorten.

Response: We have thoroughly revised the abstract to enhance its conciseness. Unnecessary details and repetitions have been removed, resulting in a more focused and compact abstract.

Comment #2: Please make sure that you explain all abbreviations when they are first mentioned in the abstract and in the main text. No need to repeat the full meaning in the text unless it is necessary. Also, it would be nice to include all abbreviations in the provided section at 326-329.

Response: In accordance with the suggestions, we have conducted a full review of the manuscript to confirm that all abbreviations are defined at first use in the abstract and main text, eliminated redundant full-term repetitions for conciseness, and updated the Abbreviations list.

Comment #3: I see that the authors present scientific hypotheses at the end of the introduction (lines 109-117) to justify the aim of this research. I think it is better to briefly explain how the study was performed to verify these hypotheses and highlight the main findings of this study, and how the results support the hypotheses. The transition from line 114 to line 115 is not logical. Do you still elaborate on the hypothesis or move to explain your findings? Please review.

Response: To test the parallel mediation hypotheses, we employed the bootstrap method with 5000 samples to generate 95% confidence intervals (CIs). The results indicated that the mediation effects were statistically significant, as the 95% CIs did not include zero, thus supporting the proposed hypotheses. A detailed elaboration of the main findings and how they substantiate the hypotheses is provided in Section 3.5. Furthermore, we have refined the logical flow between lines 107 and 111 to enhance clarity.

Comment #4: In the statistical analysis, two sentences contradict each other. The first statement indicates that all variables are normally distributed (lines 174-175). The second states that the scale scores are not normally distributed, as indicated by the Kolmogorov–Smirnov test (lines 178-179). Pearson’s correlation was used (lines 179–180), although the data were previously described as non-normally distributed. If the scale scores indeed deviate from normality, Spearman’s rank correlation would generally be more appropriate. However, Pearson’s correlation can still be valid with large sample sizes or when deviations from normality are mild. Please clarify this discrepancy and justify the use of Pearson’s correlation.

Response: The original phrasing may have been ambiguous. Our intended meaning was that all variables were tested for normality. To prevent any potential misunderstanding, we have revised the relevant descriptions in the manuscript. [Page 8, line 166-167]

Comment #5: Very limited information is provided on how the mediation analysis was performed. It refers to the model used for parallel mediation, as Model 4 at line 230, but it also stated as Model 4.2 at line 182. I am wondering whether the parallel mediation model proposed by Preacher and Hayes (2008) (line 230), DOI:10.3758/BRM.40.3.879, using the PROCESS feature (line 182) was used. Then, please include a proper reference. Also, no explanation of the model’s equations is provided (e.g., direct and indirect paths, meaning of c, c′, a₁b₁, a₂b₂ in your model).

Response: As outlined in the manuscript, we have provided a detailed description of the mediation analysis procedure. [line 172-178; line 225-227]

Specifically, the parallel mediation model has been uniformly designated as Model 4.2, and relevant references have been updated accordingly. [line 223]

Additionally, we have explained the direct and indirect pathways in the model equations, along with the interpretations of the coefficients c, c′, a₁b₁, and a₂b₂. [line 231-235]

Comment #6: At lines 195, 203, 463, I think there is a typo, where the total number is 2,317 (lines 21, 126), not 2,137. Please check.

Response: We thank the reviewer for pointing out this typographical error. The correct total number should indeed be 2,317, not 2,137. This was a clerical error that has now been corrected in the manuscript at lines 185, 193, and 465. We have also rechecked the entire manuscript to ensure no similar errors exist. We appreciate the reviewer's careful reading and valuable comments.

Comment #7: For mediation analysis and values, I noticed some discrepancies appeared between the text (lines 236-241, Table 4, and Figure 2 regarding the values of the paths. For example, at line 237 and figure 2, the value for path c’ is 0.03145; however, in Table 4 the value is 0.0325. At line 238, the value for path a1b1 is 0.0130, while in Table 4 it is 0.0128. same issue for a2b2. In Table 4, I would prefer to specify the paths corresponding to each row as in Figure 2. Please verify, check and unify the decimal throughout the manuscript.

Response: The path coefficient c′ in Figure 1 has been corrected to 0.0325, and corresponding revisions have been made both in the figure and the relevant sections of the text.

Comment #8: In Tables 1 and 2, please specify whether the smartphone usage duration is measured per day or in another way. And whether every use is counted, including for learning purposes.

Response: The smartphone use duration in Tables 1 and 2 is expressed on a daily basis and has been revised accordingly. It should be noted that the study was not designed to capture a detailed account of all specific usage behaviors. However, your comment is very insightful. We did indeed overlook the time students spend learning on their phones during our investigation.

Comment #9: Figure related: The legend of Figure 1 contains very limited information. I would rather expand some information about the abbreviations, meaning of the “+”, “-”, “M1”, “M2”, “X”, “Y”, and briefly explain your logical flow and proposed hypothesis. Same issue for Figure 2, no explanation for direct, indirect paths, meaning a₁, b₁, a₂, b₂.

Response: We have expanded the abbreviations related to the mediation analysis model in the manuscript to include the full terminology.

Comment #10: Given that the results showed that smartphone addiction is associated with poorer sleep quality. And the perceived stress and lifestyle mediate the relationship statistically. Additionally, given the study's limitations, it is not possible to infer causal relationships. However, I am missing information on whether smartphone addiction could lead to poor sleep quality because it increases stress and decreases healthy lifestyles, or what exactly the potential mechanisms for this association.

Response: We sincerely thank the reviewer for this insightful comment. We agree that a deeper theoretical discussion of the potential mechanisms underlying these statistical associations is crucial. In response, we have thoroughly revised the Discussion section to elaborate on the potential causal pathways by which smartphone addiction might impair sleep quality through increased perceived stress and reduced engagement in health-promoting behaviors.

Once again, thank you very much.

Best wishes,

Dehui Yin

---

## [Decision Letter · Decision Letter 1]

29 Dec 2025

The Influence of Smartphone Addiction on Sleep Quality Among College Students: The Parallel Mediating Roles of Perceived Stress and Health-Promoting Lifestyle

PONE-D-25-36906R1

Dear Dr. Yin,

We’re pleased to inform you that your manuscript has been judged scientifically suitable for publication and will be formally accepted for publication once it meets all outstanding technical requirements.

Kind regards,

Pavle Randjelovic, Ph.D.

Academic Editor

PLOS One

Additional Editor Comments (optional):

Reviewers' comments:

Reviewer's Responses to Questions

**Comments to the Author**

Reviewer #1: All comments have been addressed

Reviewer #2: All comments have been addressed

2. Is the manuscript technically sound, and do the data support the conclusions?

Reviewer #1: Yes

Reviewer #2: Yes

3. Has the statistical analysis been performed appropriately and rigorously?

Reviewer #1: Yes

Reviewer #2: Yes

4. Have the authors made all data underlying the findings in their manuscript fully available?

Reviewer #1: Yes

Reviewer #2: Yes

5. Is the manuscript presented in an intelligible fashion and written in standard English?

Reviewer #1: Yes

Reviewer #2: Yes

Reviewer #1: Thanks the authors for making the corresponding responses and revisions. I have no further comments and recommend acceptence of this work to be published on Plos One.

Reviewer #2: I do not have any further comments. The authors have significantly improved the manuscript. In my opinion, all my concerns have been addressed.

**Do you want your identity to be public for this peer review?** For information about this choice, including consent withdrawal, please see our Privacy Policy

Reviewer #1: No

Reviewer #2: No

---

## [Editor Report · Acceptance letter]

PONE-D-25-36906R1

PLOS One

Dear Dr. Yin,

I'm pleased to inform you that your manuscript has been deemed suitable for publication in PLOS One. Congratulations! Your manuscript is now being handed over to our production team.

Kind regards,

on behalf of

Dr. Pavle Randjelovic

Academic Editor

PLOS One